# Are teachers meeting students' needs in untracked science classrooms? Evidence based on a causal inferential approach

Eric Ho *, Michael Seltzer, Minjeong Jeon

Department of Education, University of California, Los Angeles, Los Angeles, California, United States of America

* ericmho@ucla.edu

## Abstract

Tracking has been criticized for relegating disadvantaged students to lower track courses in which students encounter a greater lack of instructional support. While an end to tracks through detracking is a possible solution, there are concerns that detracking will create more heterogeneous classrooms, making it harder for teachers to provide adequate support to their students. Using the 2015 PISA dataset, this study conducts a causal inferential analysis to understand the differences in student perceptions of teaching in tracked and untracked environments. The results provide evidence that students' needs, with respect to adaptation of instruction and provision of individualized feedback and support, are being met to the same extent on average in tracked and untracked science classes, suggesting that teachers may not necessarily have a harder time meeting the needs of students in untracked classes.

## Introduction

Are teachers meeting students' needs in untracked science classrooms and schools? Tracking is the separation of students by academic ability into different courses or curricula. These different sets of courses are called "tracks," with high tracks including courses such as Honors/Advanced Placement courses and low tracks consisting of remedial courses. In untracked schools, students are not sorted into different courses or curricula by academic ability, thereby creating classrooms with more heterogeneous levels of student ability.

Tracking is a common practice in American schools, yet the research literature has detailed equity issues related to tracking; some scholars have called for an end to tracking or "detracking" as an educational intervention to remedy such issues [1]. In this process of detracking, formerly tracked schools have removed remedial courses and promoted students in those courses to higher tracks or eliminated tracks altogether to become untracked or detracked schools [2, 3]. However, one central criticism of detracking is that with the removal of tracks that group students by ability, teachers would have a harder time meeting the needs of a classroom with more heterogeneous abilities [4, 5]. Students in detracked classrooms might feel that their teachers are not as able to meet their needs as they would be in tracked classrooms

**Data Availability Statement:** The data may be found at https://nces.ed.gov/pubsearch/pubsinfo.asp?pubid=2017120.

**Funding:** The author(s) received no specific funding for this work.

**Competing interests:** The authors have declared that no competing interests exist.

since teachers have to meet the needs of students of more varying academic abilities through adaptation of instruction and provision of adequate feedback and support.

This study aims to evaluate this criticism through causal inferential analysis by comparing student perceptions of teaching between tracked and untracked schools. The results of this study can inform policymakers and educators as to whether detracking may have unintended, negative consequences in the classroom even if it may remedy the issues created by tracking as noted by Oakes [1]. This study is also important because studies on tracking generally focus on student outcomes rather than on teaching within classrooms. Understanding how tracking or detracking affects teaching, specifically the extent to which students feel academically supported, is key in understanding why some studies have found differences in academic outcomes between tracked and detracked environments. Using the 2015 PISA dataset, we provide a cross-sectional analysis of tracked and untracked schools. Although the data did not come from a randomized experiment, through the use of propensity score weighting techniques for handling selection bias [6], along with covariance adjustment, we use the sample of untracked schools to obtain sensible estimates of the counterfactual for students in the tracked schools, in an effort to answer the question: How would the tracked students have fared in terms of the extent to which they feel academically supported had they instead been in classrooms without tracking? Do students in untracked classrooms tend to feel less academically supported than students in tracked classrooms? Might they feel more supported? Or might there be little difference?

## Equity issues in tracking and the promise of detracking

Tracking has been a relatively widespread feature of secondary school education in the United States since standardized testing became prevalent in the early 20th century [1]. Although some countries track students into vocational versus academic schools, U.S. schools tend to practice course-by-course tracking in which students are tracked within schools only with respect to certain subjects [7]. For instance, American students may be tracked into the high track of AP/Honors courses or into the low track of remedial courses for science, but not for other subjects like physical education or music. The decision to track or not depends on a school's resources and student demographics as well as characteristics of its surrounding community [1, 3, 5, 8]. In general, detracking policies have been more likely to be adopted in urban school systems with large proportions of low-income and Black and Hispanic students. The reform has been much less welcome among more affluent suburban school systems with predominantly White and Asian American student populations [5, 8].

Some educators justify the practice of tracking by claiming that tracking allows for differentiated instruction, allowing teachers to better meet the needs of a diverse student population. They argue that classes with heterogeneous levels of academic ability would cause teachers to "teach to the middle," thereby shortchanging high-achieving students by diluting the curricula [4, 5] and preventing teachers from effectively adapting their instruction and providing sufficient individual support and feedback to accommodate students of different abilities. Others believe that tracking is more appropriate for subjects like math; because concepts must be covered sequentially, tracking is necessary to fill gaps in students' understanding [9]. However, Oakes [1] discovered that teachers in low-track classes spent less time on instruction than those in high-track classes and that students in low-track classes (who are disproportionately low-income or minority students) were subject to harsher disciplinary measures, lowered expectations, and ineffectual teaching practices. Furthermore, she argued that tracking protects White middle- and upper-class privilege, is underpinned by normative prejudices around race and class, and functions to perpetuate socioeconomic inequality.

Detracking has emerged as a potential solution to address equity issues caused by tracking. Prior research on detracking has focused primarily on students' academic outcomes [2]. There is some evidence that detracking may be detrimental to the academic development of high-achieving students. For example, a study that focused on a policy abolishing remedial math classes in Chicago public high schools found that attending a class with lower-skilled peers reduced math achievement for high achievers [10]. On the other hand, detracking proponents argue that since tracking has traditionally underserved low-achieving students, it is important that detracking be implemented in efforts to help such students. Other studies have found evidence that detracking may be beneficial for lower-achieving students' achievement [11]. In a meta-analysis of over 300 studies, Hattie [12] found that tracking "has minimal effects on learning outcomes and profound negative equity effects." While the effects of tracking on academic outcomes could be mixed or negligible, the "negative equity effects," as could be captured by non-academic outcomes, warrant further attention. As noted by Oakes [1], the way teachers interact with their students can affect not only their academic outcomes but their self-efficacy and perception of what they can accomplish. Thus, a focus on student perceptions of teaching with regards to tracking and detracking seems appropriate, especially with concerns that more heterogeneous classrooms would be harder for teachers to manage.

Unfortunately, scholarly literature on teaching practices in untracked environments is sparse and usually focused on content areas other than science [13], such as English and history [14] or mathematics [9, 15]. Most of those studies of instructional practices are also qualitative, highlighting instructional practices in small samples without looking at the larger population of untracked/detracked schools. Rubin and Noguera [16] noted that there is a need for research to investigate whether foundational and complex subject areas like math or science can be effectively detracked. Watanabe et al.'s qualitative study [13] of successful instructional practices in chemistry classrooms suggested that science classes can be effectively detracked but noted that more large-scale studies need to be done.

## Purpose and goals of the study

The main purpose of this study is to provide an in-depth look into how student perceptions of instructional practices vary between untracked and tracked science class environments. We examined how not-using-tracking can potentially impact students' perceptions of teaching in science classrooms in terms of three outcomes related to the extent to which perceived teacher practices address students' needs: Student perceptions of teachers adapting instruction to their diverse needs, providing individualized support, and providing adequate feedback. Utilizing key concepts and techniques in the causal inference literature, the study advances our knowledge of the effects of detracking through the analysis of a large-scale dataset and helps us better understand students' perceptions of frequency of certain instructional practices in both untracked and tracked science classrooms. These student perceptions of teaching quality have been shown to provide reliable and valid information regarding teaching effectiveness [17–20]. Through our analysis of cross-sectional data, our comparisons of tracked and untracked classrooms allow us to evaluate a central criticism of detracking, i.e., that it would prevent teachers from meeting the needs of a heterogeneous classroom.

As discussed in the Methods section, even with data collected from a non-experimental design, we can build an evidentiary warrant for making causal claims when we are able to control for potential confounders that could bias our estimates of the effects of interest. Our assumption is that, conditional on the key confounders we have controlled for based on the findings in the previous literature, we can consider tracking as if it were randomly assigned to units. Under this assumption, which is termed the strong ignorability assumption [21], we

can, with proper weighting of units in our sample and other adjustments (e.g., ANCOVA), obtain sensible estimates of the effects of interest. While we do not have a sample of schools that have detracked (i.e. schools that were tracked but then became untracked), the students in untracked schools can serve as a source of counterfactual information, and help us understand how students in the schools with tracking would have fared in terms of their instructional needs being met, had they instead experienced detracked classes. Furthermore, since it is difficult to control for all potential confounders, we can conduct a sensitivity analysis to address the following question: If there were a potentially important confounder that was not measured in the PISA survey, how strong a confounder would that need to be to appreciably alter our results?

The goals of this study are twofold:

1. To understand how detracking affects student perceptions of how well teachers are meeting their needs by adapting instruction to diverse needs, and providing individualized support and feedback

2. To conduct sensitivity analyses to ascertain the robustness of the estimated causal effects to unobserved confounding. We also compare our key results, which are based on a strategy for handling selection bias developed by Hong [6] discussed in the next section, to results based on alternative ways of handling selection bias.

Other studies have shown the promise of detracking and guided our choice of outcomes. Our conceptual framework is largely based on underlying principles found in successfully detracked English classes as articulated by Freedman et al. [22]. They argued that best practices in detracked classrooms include "providing support to individuals as needed" and "providing opportunities for diverse ways of learning." Even though their study focused on English instruction, Rubin [23] noted that this framework has been applied to various other subjects. We utilized this framework in choosing our outcomes of interest which respectively correspond to those best practices (perceived feedback, teacher support, and adaptation of instruction). Our study investigates whether the untracked science classrooms embody those principles through analysis of students' perceptions of instructional practices. Our desire to conduct causal inferential analyses was inspired by other scholars who suggested that schools attempting to detrack tend to focus more on equalizing access to curriculum for all students, maintain higher expectations for low-track students, and improve the quality of student work [24, 25].

Therefore, given the promising effects of detracking from Freedman et al. [22] and other researchers, we hypothesized that students should perceive that their teachers are meeting their needs to roughly the same extent between tracked and untracked schools. In other words, rather than expecting statistically significant differences in the outcomes, we expected to find null effects between the two types of schools. Although teachers may find it more difficult to support students in a heterogeneous classroom, the two studies suggest that those teachers are able to adapt. This would suggest that the common criticism of detracking—that teachers are not able to meet the needs of students in more heterogeneous classrooms—is largely unfounded at least with respect to student perceptions of teaching.

## Method

### Sample

For this study we used data from the 2015 Programme for International Student Assessment (PISA) [26]. PISA is administered to 15-year-olds in schools worldwide every three years. The

2015 PISA administration focused on science assessment and included background questionnaire items related to science instruction. Considering that our research questions are centered around the effects of detracking in the U.S. educational context, our analysis was restricted to U.S. school and student data from the 2015 PISA. There were a total of 3,936 U.S. students and 137 U.S. schools.

In a quasi-experimental design, it is especially important to have a rich set of covariates to account for confounding and more accurately estimate causal effects. We used these data because important student-level and school-level covariates were measured as well as our outcomes of interest (student perceptions about adaptation of instruction, individualized support, and provision of adequate feedback by their science teachers), and a school-level item related to tracking policy that we employed as the tracking indicator. Other nationally representative datasets, such as the Education Longitudinal Study of 2002 (ELS:2002) do not offer a set of covariates and outcomes as rich as those in the PISA 2015 data; for example, in ELS:2002, variables related to teaching quality are only assessed with one or two items, and the focus is on English language arts or mathematics instruction. Furthermore, when this study was conducted, the 2015 PISA administration was the most recent PISA administration which focused on science, a subject not commonly considered in studies on tracking. Thus, unlike other datasets which do not offer such a rich set of covariates or outcome measures, this PISA dataset is a unique large-scale dataset to study the detracking effects on perceived teaching in the U.S.

**Tracking.**  There is only one variable related to school-level tracking in the PISA dataset. This variable has been used in prior studies using PISA data investigating the effects of tracking, such as in Cordero et al. [27] and in Trinidad and King [28]. The tracking variable was obtained based on a school-level item from the principal questionnaire that asked how classes at the school are grouped by ability. Possible responses to this item were (A) "for all subjects," (B) "for some subjects," and (C) "not for any subjects." Two groups are compared: the schools that selected (A) and (B), and the schools that chose (C). Group C comprises 25 schools. Group A that grouped by ability "for all subjects" includes 12 schools and group B that grouped by ability "for some subjects" includes 100 schools. We considered only groups A and C in the main analysis for a total of 910 students and 32 schools because we could not be certain that group B schools, which tracked "for some subjects," necessarily tracked for the subject of science. Thus, to be conservative and to avoid estimating inaccurate causal effects by including schools that may not have tracked science classes, we excluded group B in the main analysis. Group B is reincorporated in the sensitivity analysis which can be found in S3 Appendix.

## Outcomes of interest

Our outcomes of interest are three derived variables about students' perceptions about their science teachers' instructional practices: adaptation of instruction to diverse student needs (adaptive instruction), provision of individualized support (emotional support), and provision of adequate feedback (personal feedback) The original variable labels in the PISA codebook for these outcomes are ADINST, TEACHSUP, and PERFEED respectively. Adaptation of instruction describes how often the teacher adapts the lesson or changes the structure of a lesson to meet students' needs. Provision of individualized support refers to the amount of help provided by the teacher and their level of interest in individual students' learning. Provision of adequate feedback encapsulates the amount of feedback and advice given to students. These outcomes in the context of PISA 2015 are important because there is evidence to suggest that adaptive teaching and teacher feedback are related with science performance [29]. Furthermore, these outcomes have been shown to capture meaningful variation in teaching quality and to correlate with students' intrinsic motivations in terms of predictive validity [30].

**Table 1. Items corresponding to each outcome.**

| Outcome | Item |
|---|---|
| Adaptive instruction | The teacher adapts the lesson to my class's needs and knowledge. |
| | The teacher provides individual help when a student has difficulties understanding a topic or task. |
| | The teacher changes the structure of the lesson on a topic that most students find difficult to understand. |
| Emotional support | The teacher shows an interest in every student's learning. |
| | The teacher gives extra help when students need it. |
| | The teacher helps students with their learning. |
| | The teacher continues teaching until the student understands. |
| | The teacher gives students an opportunity to express opinions. |
| Personal feedback | The teacher tells me how I am performing in this course. |
| | The teacher gives me feedback on my strengths in this subject. |
| | The teacher tells me in which areas I can still improve. |
| | The teacher tells me how I can improve my performance. |
| | The teacher advises me on how to reach my learning goals. |

Additionally, the use of these outcomes is justified based on their strong internal consistencies, with Cronbach's alpha of 0.918, 0.944, and 0.833 for TEACHSUP, PERFEED, and ADINST respectively [26]. Therefore, focusing on these outcomes not only addresses a common criticism of detracking but also sheds light on important mediators which strongly relate with students' academic and non-academic outcomes. The items corresponding to each construct are displayed in Table 1.

These derived variables are indices that correspond to meaningful latent constructs. Each derived variable was computed from three to five Likert items from the student questionnaire. These derived variables were constructed using item response theory (IRT) scaling methodology, and the procedures used to establish construct validity were established in the PISA 2015 Technical Report [26]. See S1 Appendix for descriptive statistics of these outcome measures across the entire sample and by subgroups of students.

**Covariates.** The descriptive statistics of the school-level and student-level covariates included in the analyses are outlined in Table 2 along with their corresponding labels in the PISA school and student codebooks. We accounted for these covariates in our analyses because as mentioned earlier, the decision to track or not depends on a school's resources and student demographics: detracking policies are more popular with schools serving lower-

**Table 2. Means and standard deviations (in parentheses) of covariates between types of schools.**

| Covariate | Label in codebook | Tracked schools | Untracked schools |
|---|---|---|---|
| Percentage of students receiving free- and reduced-price lunch | FRPL | 3.4 (1.2) | 3.7 (1.1) |
| Student-teacher ratio | STRATIO | 15.6 (3.3) | 15.8 (5.7) |
| Percentage of 10th graders who are non-native English speakers | SC048Q01NA | 24.6 (27) | 27.1 (31) |
| Percentage of 10th graders from SES-disadvantaged homes | SC048Q03NA | 46.2 (29.9) | 54.2 (27.9) |
| Percentage of 10th grade students enrolled in special education programs | SC048Q02NA | 18.9 (10.4) | 14.1 (8.9) |
| Resources available for science instruction | SCIERES | 6.5 (1.3) | 5.1 (2.2) |
| Levels of economic-socio-cultural status | SC048Q02NA | 0.2 (0.5) | -0.1 (0.6) |

The only student-level covariate is "levels of economic-socio-cultural status." All other variables are school-level covariates.

income, disadvantaged populations while tracking remains popular with more affluent suburban school systems. The covariates thus include various measures that relate to school resources and student demographics to control for confounding stemming from pre-existing differences between the tracked and untracked schools that could influence the outcomes independently of the decision to track or not.

Resources available for science instruction (SCIERES) and school mean economic-socio-cultural status (ESCS) were chosen as the important covariates for the sensitivity analyses because we believe they are the strongest observed confounders in the data. Schools with high mean achievement that serve more economically-advantaged students are more likely to track [3, 8, 31]. Such schools are likely to have more resources for science instruction than do schools serving less affluent communities. Furthermore, levels of resources for science instruction should correlate positively with how well science teachers meet the needs of their students, especially since SCIERES includes indicators of teacher experience and expertise. For those reasons, we believe ESCS and SCIERES positively influence both a school's propensity to track as well as the student-teacher responsiveness outcomes, thus making them suitable benchmark covariates as confounders to be accounted for. Prior research also suggests that cultural and class differences between poor, Black, and/or Hispanic students and teachers who are mostly White and socioeconomically advantaged can impede student-teacher relations [32–34]. Conversely, we expected more affluent students at predominantly White and Asian American schools to feel more comfortable asserting their needs and to perceive their teachers as sympathetic and responsive.

The FRPL and SCIERES variables were coded in the PISA dataset as ordinal variables, with higher numbers corresponding to higher percentages of students receiving free/reduced price lunch and more resources available for science instruction respectively. Schools that do not track appear on average to have fewer resources for science instruction and higher percentages of students receiving free/reduced-price lunch and socioeconomically disadvantaged students than do schools that do track, thereby supporting the findings in the research literature.

**Additional remarks regarding our approach.**   Although the number of covariates may seem limited, we believe that we have incorporated the major confounders that might pose problematic in accurately estimating the causal effects. Following the advice of Steiner et al. [35], rather than blindly incorporating many different covariates (which would cause estimation problems given the rather small sample size), we deliberately aimed for constructs deemed in the literature to be highly correlated with our outcome and tracking and incorporated multiple measures of those constructs. In doing so, we believe that we have incorporated the major confounders in our analyses that could potentially result in biased causal effects if not controlled for. For example, given that schools in disadvantaged communities are less likely to track, we attempted to encapsulate this construct of "disadvantage" using proportions of the school population that receive special education services, that are non-native English speakers, and that are from low-SES households. Furthermore, we strove to include mostly demographic variables as covariates since we did not want to control for covariates that could be affected by being in a tracked or untracked classroom, such as student academic achievement (which may be affected by being in tracked or untracked classrooms as previous studies have shown). Including such variables would introduce post-treatment selection bias [36] and control away possible differences in the outcomes between the tracked and untracked groups.

## Description of the approach

Estimating the causal effect of detracking can be challenging with non-experimental multistage survey data that has a multilevel structure. Thus in the first stage of our analysis, we first

computed the appropriate weights for the units in our study to ensure balance in pre-treatment covariates between the tracked and untracked groups of students (we call these groups our treatment and control groups respectively for simplicity even though these terms are generally reserved for strictly experimental designs), while also accounting for the survey weights. To account for the multilevel structure of the data, we incorporated the weights into a multilevel model to estimate the average treatment effects.

Another significant challenge in causal inference is establishing that all key confounders have been properly accounted for and providing evidence of the robustness of the results. As such, in the second stage of our analyses, we conducted sensitivity analyses which help investigate the robustness of our results by considering the effects of unobserved confounders. We can determine how much unobserved confounding would have to exist to nullify our findings. While we controlled for a set of key covariates related to the outcome of interest and to whether a school has tracking in all subject areas or no tracking, there very well could be another potential confounder that was not measured in the PISA survey and thus cannot be controlled for. By conducting a sensitivity analysis, we can assess how robust our results might be to an unmeasured confounder. That is, we can conduct an analysis that answers the following question: If there were a potentially important confounder that was not measured in the PISA survey, how strong a confounder would that need to be to appreciably alter our results? If it would take an unusually strong confounder to alter our results—a confounder that is substantially stronger than the strongest confounder in our data set—that would give us more confidence in our results. Additionally, we also compared the different results obtained using different methods, such as those obtained from a naïve difference-in-means estimate.

**Marginal mean weighting.** In efforts to handle selection bias, we adopted as our estimation strategy the marginal mean weighting through stratification approach (MMW-S) developed by Hong [6], which is suitable for multilevel data (e.g., students nested in different schools), and combines the strengths of propensity score stratification and weighting. This approach has various advantages. Firstly, it provides a valuable approach to estimating causal effects from observational, nonrandomized data, as Hong did with her approach to investigate the effects of heterogeneous classroom groupings. Secondly, although other weighting schemes such as inverse-propensity-of-treatment weighting (IPTW) exist (see, e.g., [37], the MMW-S approach is more robust to misspecifications of the propensity score model compared to those approaches. Finally, the MMW-S approach avoids additional bias by assigning zero weights to units that have no proper counterfactuals due to lack of common overlap in the covariate distributions between the treatment groups (which is not true for IPTW). Our approach is similar to multilevel matching, in which the balance of covariates is established before using a regression model to correct for any additional bias. Such an approach is appropriate for clustered observational studies as we see here [38].

To apply this approach, we first created student-level propensity scores based on the school-level and student-level covariates. These propensity scores are useful for causal inference in balancing the distributions of covariates [21]. These propensity scores were used to create strata of students, each of which contains students that are otherwise similar across important covariates but differ in whether they were tracked or not. The proportions of students in each treatment group and stratum are used to create marginal mean weights which weight the students such that the no-tracking sample resembles the tracking samples.

For a given student in propensity score stratum s and treatment group z, the marginal mean weight (MMW) for that student's outcome $Y$ is calculated as $\frac{n_s}{n_{z,s}} \times pr(Z = z)$ where $n_s$ is the number of students in stratum $s$, $n_{z,s}$ is the number of students in stratum $s$ who received treatment $z$, and $pr(Z = z)$ is the proportion of students in the whole sample who received

treatment $z$. Intuitively, these MMWs function similarly to inverse propensity scores—due to self-selection, certain students may be overrepresented or underrepresented in treatment groups, thereby biasing the estimated treatment effect for the population of interest. These weights adjust for this by upweighting students in treatment groups who are underrepresented in certain strata (i.e. smaller denominator) and downweighting those who are overrepresented (i.e. larger denominator).

Causal effects can be estimated from these marginal mean weights because they function similarly to inverse propensity score weights in ensuring that the distribution of covariates is similar between the treatment and control groups. The weights created from this approach are important because they allow us to properly adjust for selection bias, thereby more accurately evaluating the effects of detracking on the student perceptions on teaching in science classrooms. A similar approach was also used by Jiang and McComas [39] in their PISA-based study which is largely replicated here in this analysis. After the weights are calculated, more accurate estimates of the standard errors of the treatment effects can be obtained from incorporating them into a multilevel model as was done by Hong [6].

**Common support.**   A logistic regression model was used as the propensity score model with the log-odds of a student experiencing tracking in science versus not experiencing tracking modeled as a function of the set of covariates discussed above. Based on the fitted model, the expected log-odds of experiencing tracking was obtained for each student. These expected values constitute the propensity scores used in the MMW-S approach described above to obtain strata of students, with each stratum consisting of a group of students who experienced tracking and a group that did not, who were similar (i.e., balanced) with respect to the covariates used to construct the propensity scores.

To ensure that causal inferences are not biased by cases with no potential counterfactual cases, it is important to find the region of common support which contains students who are comparable in terms of their covariates, i.e. students who have comparable counterparts in the other group. As suggested by Hong [40], this region was found by only including students between two cutoffs—the maximum of the minimum propensity scores (i.e. the smallest propensity scores in the treatment and control groups) and the minimum of the maximum propensity scores (i.e. the largest propensity scores in the treatment and control groups).

Trimming observations outside the region of common support may be problematic because this leads to more missing data; see S2 Appendix for more details.

**Stratification.**   We stratified the trimmed dataset into six different strata based on the covariates, using the MatchIt package in R [41]. The number of strata was chosen to ensure that at least five strata were created to reduce most of the selection bias [42] while ensuring that no stratum was too sparse from creating too many strata. Even before the marginal mean weights were calculated, balance among the covariates within each stratum (and overall across strata) appeared to be satisfactory. The marginal mean weight for each student was calculated based on the student's membership in each stratum and the number of students in that stratum.

It may seem unusual that the MMW were calculated at the student-level even though the treatment is nominally at the school-level. This is because in an email from Professor G. Hong in the Department of Comparative Human Development at the University of Chicago (written communication, June 11, 2019) "sometimes even though a treatment is seemingly at the organization level, treatment selection may occur at the individual level if individuals select treatments by selecting the organizations." Certainly students (usually by virtue of their parents) self-select into certain schools. As mentioned before, student populations vary between tracked and untracked schools, with wealthier students finding themselves in tracked schools, perhaps because of the effectively maintained inequality hypothesis [3]. Parents with high economic,

social, and cultural capital may be more likely to enroll their children in tracked schools where they can effectively maintain their societal position. Thus, it would make sense to calculate weights at the student-level. School-level and student-level covariates balance was achieved after weighting, suggesting that this across-cluster matching, which requires proper inclusion and modeling of the between-cluster covariates, is appropriate in this situation [43].

**Incorporation of survey weights.** The literature on how to properly incorporate survey weights from complex, multistage surveys like PISA for causal inference is rather sparse, and solving this issue remains an open question [44]. Therefore, this study used the same procedure as used by Jiang and McComas [39] to combine the MMW and the student survey weights. These weights were multiplied together to yield final weights for each student, which were then incorporated into the multilevel model. The goal was to ensure balance of covariates between the treatment and control groups using the MMW while also accounting for the sampling procedure used by PISA to ensure the proper representation of over- or under-sampled groups of students. For example, in the PISA data collection, a given student may be under-sampled because of non-response bias from that student's demographic. To ensure that the student represents the correct number of students in the target population, the survey weight for that student should be higher. Also, that student may be underrepresented in the treatment group due to the self-selection process. That student's relatively high MMW should be combined with the survey weight. Thus, to compute accurate estimates of the treatment effect, which requires that the analytic survey sample represents the target population and that self-selection into treatment groups is accounted for, it is necessary to incorporate both weights.

**Balance of covariates.** For every student, each covariate was weighted by the computed final weights (i.e. the combination of the survey weight and MMW), and the absolute standardized bias between the two groups was computed by taking the difference between the mean covariate values in each treatment group and dividing it by the standard deviation of the treatment group. This measure denotes the difference in the mean covariate values in standard deviation units. Stuart [45] suggested that the absolute standardized bias for each covariate should not exceed 0.25 to prevent bias in treatment effect estimates that stem from large-enough imbalances in covariates. Table 3 shows that although some imbalances remain after incorporating the MMW, none of the absolute standardized bias values exceed 0.25, suggesting that balance of covariates between the two groups was largely achieved, even when survey

**Table 3. Absolute standardized biases under different weighting schemes.**

| Covariate | No weights | Only survey weights | MMW and survey weights |
|---|---|---|---|
| Percentage of students receiving free- and reduced-price lunch (FRPL) | 0.11 | 0.10 | 0.02 |
| Student-teacher ratio (STRATIO) | 0.24 | 0.08 | 0.02 |
| Percentage of 10th graders who are non-native English speakers (No_Eng) | 0.00 | 0.14 | 0.15 |
| Percentage of 10th graders from SES-disadvantaged homes (SES) | 0.12 | 0.11 | 0.04 |
| Percentage of 10th grade students enrolled in special education programs (SPED) | 0.08 | 0.08 | 0.05 |
| Resources available for science instruction (SCIERES) | 0.25 | 0.11 | 0.04 |
| Levels of economic-socio-cultural status (ESCS) | 0.20 | 0.19 | 0.06 |

The only student-level covariate is "levels of economic-socio-cultural status." All other variables are school-level covariates.

weights were used in computing the final weights for each student. The absolute standardized biases were also computed for the trimmed sample without any weights, with only the survey weights, and with the final weights as computed by combining the MMW and the survey weights. Results in Table 3 show that the final weights in the rightmost column of the table achieved the lowest absolute standardized biases for five out of the seven covariates, suggesting that the aforementioned procedure was most successful in improving balance of covariates between the two groups.

The line plots in Fig 1 also tell the same story by visualizing the absolute standardized biases in Table 3. For most of the covariates, the combination of MMW and survey weights was most effective (except for a few covariates) in reducing the absolute standardized bias compared to not using weights at all or only the survey weights. Our weighting scheme appears to be appropriate.

**Incorporating the MMW into the multilevel model.** Along the same line of reasoning with Hong [6], the MMW created from those strata combined with the PISA-derived survey weights, (i.e. the final weights) were then incorporated into our multilevel model. A

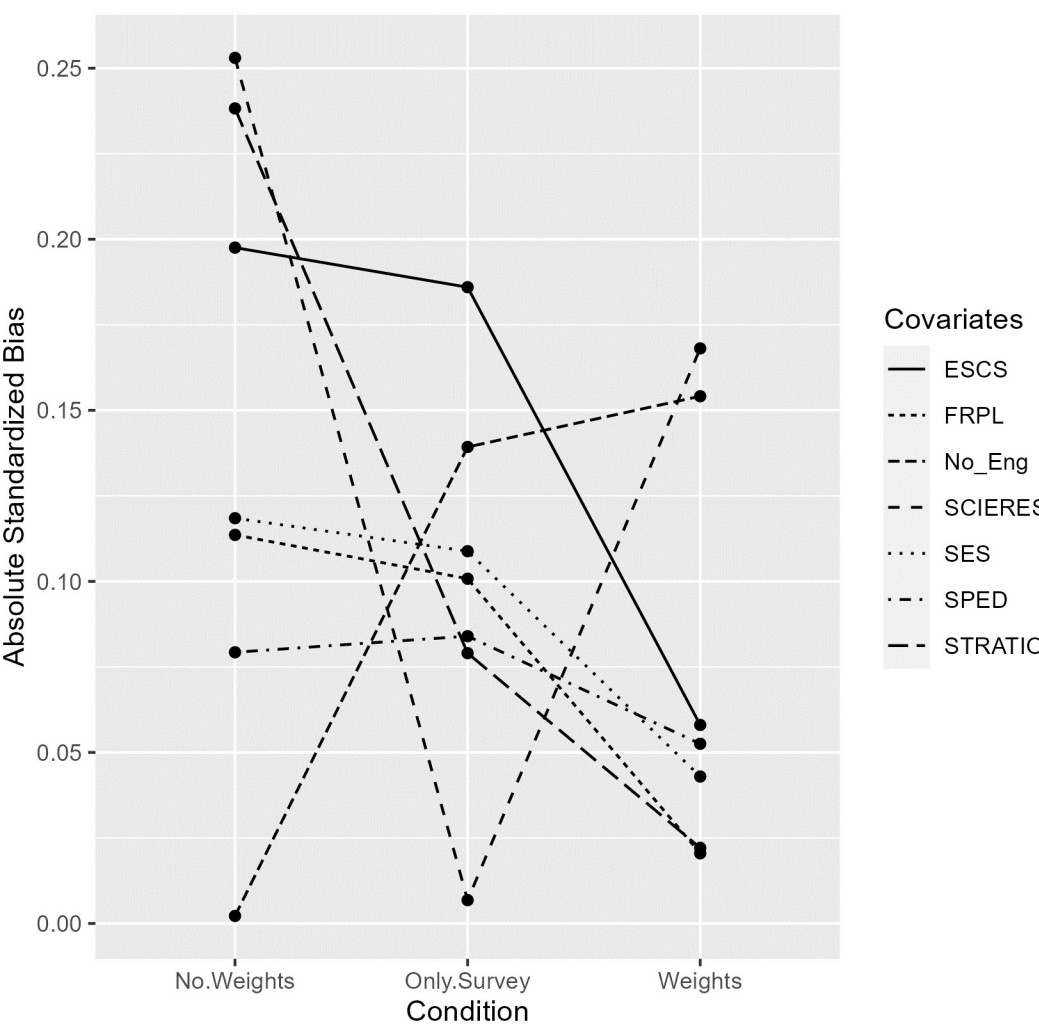

**Fig 1. Absolute standardized bias under different weighting schemes.**

multilevel analysis acknowledges the multilevel structure of the data and yields correct standard errors of the estimates. We used a two-level, random intercepts model with school-level random effects. Level 1 of the multilevel model was at the student-level, which included the student-level outcomes. Level 2 of the model was at the school-level, which included a treatment indicator variable to denote a tracked or non-tracked school. The model can be specified as

$$Y_{ij} = \gamma_0 + \gamma_1 Z_j + \mu_j + e_{ij}, \tag{1}$$

where $Y_{ij}$ is the outcome (adaptive instruction, personal feedback, or emotional support) for student $i$ attending school $j$, and $u_j$ and $e_{ij}$ are school-specific and student-specific random effects, respectively. Estimating the coefficient $\gamma_1$ for the treatment indicator $Z_j$ yielded the estimated treatment effect along with the proper standard error using the lme4 package [46].

## Results

The estimated effect of tracking versus non-tracking for adaptive instruction is -0.01663 ($SE = 0.08705$, $p = 0.85$); for personal feedback, 0.06242 ($SE = 0.10038$, $p = 0.54$); for emotional support, 0.04468 ($SE = 0.09602$, $p = 0.65$). These results are presented in the original units as provided by PISA; the outcome variables are latent variable scores that have been standardized with mean of zero and standard deviation of one. For all three outcomes, the treatment effects were not statistically significant from zero at the $\alpha = 0.1$ level. These results suggest that student perceptions of these outcomes do not differ significantly across tracked and untracked science classrooms.

These results confirm our hypothesis that detracking should not have significant adverse effects on teachers' abilities to meet their students' needs. The average levels of student perceptions of teachers' abilities to adapt instruction and provide personalized feedback and support appear to be roughly similar across the tracked and untracked schools. Because of our strong ignorability identification assumption, supported by the fact that we controlled for the key confounders based on the previous literature, we are confident that the untracked schools are reasonable counterfactuals for the tracked schools, thus giving us an accurate picture of the outcomes that we would have observed had the tracked schools decided to detrack.

## Sensitivity analyses

Following the causal inferential analysis are the sensitivity analyses. There is still the possibility that a certain amount of bias might remain due to unmeasured variables that have not been controlled for. Using the sensitivity analysis statistics and software from Cinelli and Hazlett [47], we conducted a sensitivity analysis that helps us ascertain how strong an unmeasured confounder would need to be to appreciably alter our conclusions. This analysis combined schools that track for all or some subjects and compared them with schools that do not track for any subjects. The results of the sensitivity analyses show that the resulting effect estimates are like those found in our main analyses and suggest that our results are consistent with those estimated using other methods. Most importantly, they also suggest that it would take an implausibly strong confounder—a confounder more than five times as strongly correlated with the tracking variable and outcome as the benchmark covariates of FRPL and SCIERES— to alter the conclusions drawn from the causal inferential analysis. We also assessed the differences in results based on different methods used to estimate the causal effects (such as the naïve difference-in-means estimate). See S3 Appendix for more details.

## Within-school relationships of SES with outcomes

It is also worth considering the within-school relationship between student socioeconomic status (SES; considered here using ESCS) and our three key outcomes. Within a given school, the outcomes reported by the students may vary based on the students' levels of ESCS. These can be measured by within-school slopes relating ESCS to the outcomes. The main question is, do these slopes vary across schools, specifically between tracked and untracked schools? If so, there may be systematic differences between these schools that we have failed to account for.

To that end, we conduct a multilevel analysis with the following model

$$Y_{ij} = B_{0j} + B_{1j}(ESCS_{ij} - \overline{ESCS_j}) + e_j, e_j \ N(0, \sigma^2) \tag{2}$$

$$B_{0j} = \gamma_{00} + \gamma_{01} * TRACKING_j + U_{0j}, U_{0j} \ N(0, \tau_0) \tag{3}$$

$$B_{1j} = \gamma_{10} + \gamma_{11} * TRACKING_j + U_{1j}, U_{1j} \ N(0, \tau_1) \tag{4}$$

in which $Y_{ij}$ is the outcome score for student $i$ in school $j$, $B_{0j}$ by virtue of group-mean centering represents the mean outcome score for school $j$, $\gamma_{00}$ is the average group-mean outcome score for the population of schools, $\gamma_{01}$ represents the effect of the presence of tracking ($TRACKING_j$) on the intercept, $\tau_0$ represents how much the $B_{0j}$'s vary around the grand mean after accounting for tracking, $B_{1j}$ is the slope between ESCS and the outcome for school $j$, $\gamma_{10}$ is the average of the within-school slopes for the population of schools, $\gamma_{11}$ represents the effect of the presence of tracking ($TRACKING_j$) on the slope, and $\tau_1$ represents how much the $B_{1j}$'s vary around the average slope for the population of schools after accounting for tracking. Our main focus is on $\gamma_{01}$ and $\gamma_{11}$; if they are negligible, that shows that the tracked and untracked schools are similar on average in terms of school mean outcome scores and also in the magnitude of the within-school slopes relating differences in student ESCS to differences in outcomes.

The results of the multilevel models show that in general, for the three outcomes, those values are not statistically significant (see Table 4). Therefore, we do not have enough evidence to believe that the tracked and untracked schools we consider differ systematically in terms of their outcomes scores or in their relationships between student ESCS and outcomes. These schools are in fact comparable.

## Discussion

This study investigated whether teachers may find it harder to adapt their instruction and address student needs in more heterogeneous classes resulting from detracking. Proponents of tracking often justify the practice by asserting that teachers can better serve students' individual needs when they are grouped into classes by ability level. For example, Yonezawa and Jones [48] found that some students said their teachers were ill-prepared to teach mixed-ability classrooms. However, based on the sample of U.S. high schools in the 2015 PISA data we found no evidence that tracking has a positive effect on students' perceptions of teacher

**Table 4. Estimates and standard errors (in parentheses) of $\gamma_{01}$ and $\gamma_{11}$.**

| Outcome | $\gamma_{01}$ | $\gamma_{11}$ |
|---|---|---|
| Adaptation of instruction | -0.03 (0.09) | -0.01 (0.08) |
| Personal feedback | 0.06 (0.10) | 0.07 (0.12) |
| Emotional support | 0.05 (0.06) | 0.18 (0.08) |

responsiveness in science courses. There seems to be no evidence that students' perceptions of teacher responsiveness in science courses were related to whether the courses were tracked—detracking may not have adverse effects on students' perceptions of teaching based on the overall, average effects. None of the effect estimates related to our outcomes of interest—adaptation of instruction, provision of adequate feedback, or provision of individualized support—were statistically significantly different from zero. Thus, our study provides preliminary evidence that detracking may yet be a promising educational intervention that does not negatively affect students' perceptions of their teachers' ability to meet their students' needs.

We also believe that our results are robust to the approach used to estimate the effects and to unobserved confounding based on the results of our sensitivity analyses. We found that very strong unobserved confounding of a magnitude more than five times larger than the strongest observed confounders (resources available for science instruction and school mean economic-socio-cultural status) would not be sufficient to increase the point estimates of the effect sizes to 0.2, which is the low end of the commonly accepted "small-to-medium" effect size range in the social sciences [49]. Based on prior research findings that school mean socioeconomic status is a very strong predictor of a school's tracking policy, as well as evidence in the PISA data analyzed here that our benchmark confounders are the strongest observed predictors of tracking status and the outcomes, we believe that unobserved confounding of that magnitude is highly implausible.

## Significance and implications

This study is important for a variety of reasons. Firstly, it addressed a common criticism of detracking—that teachers would find it more difficult to meet the needs of a student body with more heterogeneous abilities—by comparing the experiences of students in tracked and untracked classrooms. The answer to this question relates to the viability of detracking as a solution for educational inequities caused by tracking. Detracking may not be desirable on a large scale if it hampers teachers' abilities to meet their students' needs. The results from this study suggest that this fear may be unfounded. Other criticisms of detracking may exist but this particular criticism is not supported by the results of this study. Additionally, this study could rule out one reason for the differences in academic outcomes found between tracked and untracked environments in other studies; if teachers are able to meet their students' needs to roughly the same extent in both kinds of environments, then there should be another reason why differences in academic outcomes exist. The results of this study have significant implications for policymakers and educators interested in reducing inequalities in youth outcomes through detracking yet who may be concerned about unintended negative consequences.

Secondly, this study used causal inferential techniques on a large-scale, nationally representative data set which has only been done with few other studies on tracking. The findings from this study are more generalizable and relevant for policymakers, rather than relying on case studies of a few schools or a single school district. Rather than relying on small case studies, policymakers can make more informed decisions based on the relatively larger sample used in our study.

The focus on teaching quality as the outcome—rather than student academic outcomes as usually found in studies on tracking—may also interest methodologists. Fu and Mehta [11] used a structural model to assess the direct and indirect impacts of tracking on student academic outcomes. They included a coefficient in the model that "allows teaching to be more or less effective in classes with widely heterogeneous students," based on the interaction of the variation of peer abilities and school effort, defined as the workload given to the class. Given the rich data on student perceptions of teaching in the PISA data, an interesting extension of

their model and methodology might include more multidimensional measures of teaching quality as was done in this study.

Finally, the study focused on non-academic outcomes in the subject of science, whereas most other studies on detracking have focused largely on academic outcomes in English or math. The non-academic outcomes of interest, namely the extent to which students perceive that their teachers are meeting their needs, deserve further consideration, especially in a foundational subject like science. The U.S. government has a vested interest in addressing minority underrepresentation in STEM [50]. Given that tracking has traditionally disadvantaged minority students and put them into classrooms unlikely to inspire critical thinking or interest in the subject matter, it is thus important to understand whether schools could better serve these students in science classrooms through detracking, or at least whether detracking might have unintended negative consequences for other students. Our study fills this important gap in the research literature on tracking and presents preliminary evidence suggesting that the whole population of students (and potentially various subgroups) would not be underserved by detracking.

## Limitations of the present study

We caution, however, that our findings have some limitations mainly due to the limited sample size. Since the standard errors of our effect size estimates were fairly large possibly due to the small sample size (which unfortunately cannot be enhanced by including other countries or timepoints, since our study focuses on American schools and PISA data is not longitudinal), we cannot completely rule out the possibility that tracking increases teacher responsiveness to a degree that would be substantively meaningful. The discussion around tracking would benefit from further analyses along these lines with larger nationally representative data sets that could offer more statistical power. This limitation prevents meaningful analysis of subgroups of interest. A larger sample size would allow for analyses within subgroups of interest. Preliminary evidence from Table S1–1 in S1 Appendix suggests that while most outcomes look roughly the same between tracked and untracked schools for certain subgroups, there could be key differences in other outcomes between tracked and untracked schools for other subgroups. Given the documented detrimental effects of tracking on disadvantaged students such as students of color or students from low-income families, it may be enlightening to further investigate how perceptions of teaching vary for those students across tracked and untracked classrooms. Nevertheless, this analysis is a useful step forward and we hope it contributes some needed clarity to the debate around tracking and inspires further work on this topic.

## Supporting information

**S1 Appendix. Descriptive statistics.** We provide descriptive statistics between tracked and untracked schools and among different student demographics.
(DOCX)

**S2 Appendix. Missing data problem.** We provide descriptive statistics of the outcomes between tracked and untracked schools and among different student demographics.
(DOCX)

**S3 Appendix. Sensitivity analyses.** We conduct sensitivity analyses to evaluate whether our results seem reasonable and robust to unobserved confounding.
(DOCX)

## Author Contributions

**Conceptualization:** Eric Ho.

**Data curation:** Eric Ho.

**Formal analysis:** Eric Ho.

**Investigation:** Eric Ho, Michael Seltzer, Minjeong Jeon.

**Methodology:** Eric Ho, Michael Seltzer, Minjeong Jeon.

**Project administration:** Eric Ho, Michael Seltzer, Minjeong Jeon.

**Software:** Eric Ho.

**Supervision:** Eric Ho, Michael Seltzer, Minjeong Jeon.

**Validation:** Eric Ho, Michael Seltzer, Minjeong Jeon.

**Visualization:** Eric Ho.

**Writing – original draft:** Eric Ho.

**Writing – review & editing:** Eric Ho, Michael Seltzer, Minjeong Jeon.

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
