## [Decision Letter · Decision Letter 0]

21 Nov 2023

PONE-D-23-30374Are teachers meeting students’ needs in untracked science classrooms? Evidence based on a causal inferential approachPLOS ONE

Dear Dr. Ho,

Thank you for submitting your manuscript to PLOS ONE. After careful consideration, we feel that it has merit but does not fully meet PLOS ONE’s publication criteria as it currently stands. Therefore, we invite you to submit a revised version of the manuscript that addresses the points raised during the review process. The points raised can be found at the end of this email.

Be sure to:Indicate which changes you require for acceptance versus which changes you recommendAddress any conflicts between the reviews so that it's clear which advice the authors should followProvide specific feedback from your evaluation of the manuscriptPlease submit your revised manuscript by Jan 05 2024 11:59PM. If you will need more time than this to complete your revisions, please reply to this message or contact the journal office at plosone@plos.org. Please include the following items when submitting your revised manuscript:A rebuttal letter that responds to each point raised by the academic editor and reviewer(s). You should upload this letter as a separate file labeled 'Response to Reviewers'.A marked-up copy of your manuscript that highlights changes made to the original version. You should upload this as a separate file labeled 'Revised Manuscript with Track Changes'.An unmarked version of your revised paper without tracked changes. You should upload this as a separate file labeled 'Manuscript'.We look forward to receiving your revised manuscript.

Kind regards,

Chinaza Uleanya

Academic Editor

PLOS ONE

Additional Editor Comments:

The author should take note of the following, revise accordingly and submit.

There are any grammatical errors should be corrected at revision.

For example (in abstract):

"Tracking has been criticized........ which students encounter greater lack of instructional support."

I suggest:

"Tracking has been criticized........ which students encounter a greater lack of instructional support."

Other example:

Line 566: The results from this study have

Line 567: strong implications and significance for policymakers and educators .........

I suggest:

Line 566: The results of this study have

Line 567: significant implications for policymakers and educators .........

Reviewers' comments:

Reviewer's Responses to Questions

**Comments to the Author**

1. Is the manuscript technically sound, and do the data support the conclusions?

Reviewer #1: No

Reviewer #2: Yes

2. Has the statistical analysis been performed appropriately and rigorously? 

Reviewer #1: No

Reviewer #2: Yes

3. Have the authors made all data underlying the findings in their manuscript fully available?

Reviewer #1: Yes

Reviewer #2: Yes

4. Is the manuscript presented in an intelligible fashion and written in standard English?

Reviewer #1: Yes

Reviewer #2: Yes

5. Review Comments to the Author

Reviewer #1: 1- The findings have some limitations mainly due to the limited sample size.

2- The standard errors of effect size estimates were fairly large

possibly due to the small sample size.

3- The study focuses on American schools.

4- The authors have the option to publish the results from this study in a more appropriate journal. The comments 1 to 3 suggest that these results only provide preliminary evidence that detracking may be a promising educational intervention that does not negatively affect students’ perceptions of their teachers' ability to meet their students' needs.

5- There are any grammatical errors should be corrected at revision.

For example (in abstract):

"Tracking has been criticized........ which students encounter greater lack of instructional support."

I suggest:

"Tracking has been criticized........ which students encounter a greater lack of instructional support."

Other example:

Line 566: The results from this study have

Line 567: strong implications and significance for policymakers and educators .........

I suggest:

Line 566: The results of this study have

Line 567: significant implications for policymakers and educators .........

Reviewer #2: 1. Evidence based on a causal inferential approach is currently applied in the educational and social settings. Moreover, it is one of the most successful epistemological breakthroughs in the field of post-pandemic research today.

2. Well, according to the logic of deductive inference, the data provided by the study support the argument used. This argument in turn guarantees and facilitates the understanding of the fact or phenomenon under study.

6. PLOS authors have the option to publish the peer review history of their article (what does this mean?). If published, this will include your full peer review and any attached files.

Reviewer #1: No

Reviewer #2: No

---

## [Author Response · Author response to Decision Letter 0]

22 Nov 2023

Dear Academic Editor and Reviewers:

Thank you for your thoughtful feedback and comments regarding our manuscript entitled Are teachers meeting students’ needs in untracked science classrooms? Evidence based on a causal inferential approach (PONE-D-23-30374). Below, we have explained the changes we have made to address your 

comments. These changes are highlighted in the revised version of the manuscript.

Reviewer 1 Comments

```

1- The findings have some limitations mainly due to the limited sample size.

2- The standard errors of effect size estimates were fairly large

possibly due to the small sample size.

3- The study focuses on American schools.

4- The authors have the option to publish the results from this study in a more appropriate journal. The comments 1 to 3 suggest that these results only provide preliminary evidence that detracking may be a promising educational intervention that does not negatively affect students’ perceptions of their teachers' ability to meet their students' needs.

```

Thank you for your comments and suggestions. Comments 1 through 3 were mentioned in the “Limitations” section. In the “Discussion” section, we have also noted that our results are only preliminary evidence regarding the promise of detracking. Finally, the authors have collectively agreed that PLOS ONE would be a suitable outlet for our study.

```

5- There are any grammatical errors should be corrected at revision.

For example (in abstract):

"Tracking has been criticized........ which students encounter greater lack of instructional support."

I suggest:

"Tracking has been criticized........ which students encounter a greater lack of instructional support."

Other example:

Line 566: The results from this study have

Line 567: strong implications and significance for policymakers and educators .........

I suggest:

Line 566: The results of this study have

Line 567: significant implications for policymakers and educators .........

```

Thank you for your comments. We have edited the text as you have suggested.

Reviewer 2 Comments

```

1. Evidence based on a causal inferential approach is currently applied in the educational and social settings. Moreover, it is one of the most successful epistemological breakthroughs in the field of post-pandemic research today.

2. Well, according to the logic of deductive inference, the data provided by the study support the argument used. This argument in turn guarantees and facilitates the understanding of the fact or phenomenon under study.

```

Thank you for your support of our manuscript.

Please do not hesitate to contact us for more information. Thank you again for your consideration of our manuscript.

Best regards,

The authors

---

## [Decision Letter · Decision Letter 1]

1 Mar 2024

Are teachers meeting students’ needs in untracked science classrooms? Evidence based on a causal inferential approach

PONE-D-23-30374R1

Dear Dr. Jeon

We’re pleased to inform you that your manuscript has been judged scientifically suitable for publication and will be formally accepted for publication once it meets all outstanding technical requirements.

Kind regards,

Chinaza Uleanya

Academic Editor

PLOS ONE

Additional Editor Comments (optional):

Congratulations you paper has been accepted for publication.

Reviewers' comments:

Reviewer's Responses to Questions

**Comments to the Author**

1. If the authors have adequately addressed your comments raised in a previous round of review and you feel that this manuscript is now acceptable for publication, you may indicate that here to bypass the “Comments to the Author” section, enter your conflict of interest statement in the “Confidential to Editor” section, and submit your "Accept" recommendation.

Reviewer #3: All comments have been addressed

2. Is the manuscript technically sound, and do the data support the conclusions?

Reviewer #3: Yes

3. Has the statistical analysis been performed appropriately and rigorously? 

Reviewer #3: Yes

4. Have the authors made all data underlying the findings in their manuscript fully available?

Reviewer #3: Yes

5. Is the manuscript presented in an intelligible fashion and written in standard English?

Reviewer #3: Yes

6. Review Comments to the Author

Reviewer #3: (No Response)

7. PLOS authors have the option to publish the peer review history of their article (what does this mean?). If published, this will include your full peer review and any attached files.

Reviewer #3: No

---

## [Editor Report · Acceptance letter]

2 Apr 2024

PONE-D-23-30374R1 

PLOS ONE

Dear Dr. Jeon, 

I'm pleased to inform you that your manuscript has been deemed suitable for publication in PLOS ONE. Congratulations! Your manuscript is now being handed over to our production team.

Kind regards, 

on behalf of

Dr. Chinaza Uleanya 

Academic Editor

PLOS ONE